# Categorizing IQOS-Related Twitter Discussions

**DOI:** 10.3390/ijerph18094836

**Published:** 2021-04-30

**Authors:** Joshua O. Barker, Julia Vassey, Julia C. Chen-Sankey, Jon-Patrick Allem, Tess Boley Cruz, Jennifer B. Unger

**Affiliations:** 1Department of Preventative Medicine, Keck School of Medicine, University of Southern California, Los Angeles, CA 90033, USA; vassey@usc.edu (J.V.); allem@usc.edu (J.-P.A.); tesscruz@usc.edu (T.B.C.); unger@usc.edu (J.B.U.); 2Division of Intramural Research, National Institute on Minority Health and Health Disparities, Bethesda, MD 20892, USA; julia.chen-sankey@nih.gov

**Keywords:** social media, IQOS, tobacco, Twitter

## Abstract

(1) Background: The heated tobacco product IQOS, by Philip Morris International, is now available in over 55 countries, including the United States. Social media sites such as Twitter are often used to promote or discuss tobacco products, though prior research has not examined how IQOS is presented on Twitter. (2) Methods: This study collected and categorized Twitter conversations involving IQOS. A manual content analysis was performed on N = 3916 English tweets related to IQOS published internationally between 1 January 2020 and 30 June 2020. (3) Results: Most tweets were either online marketing for IQOS (32.3%) or personal testimonials related to IQOS use (34.2%). Personal testimonial tweets made harm reduction claims about IQOS either as an avenue to quit smoking/tobacco use (3.4%), or in comparison to combustible cigarettes (2.0%). Tobacco policy-related tweets were detected (13.9%), split between discussions of United States (4.9%) and international (4.4%) policies. News media tweets (14.2%) were also detected. (4) Conclusions: Our study suggests IQOS may be understood as a less harmful alternative to vaping and combustible cigarettes. Discussions also suggest IQOS is likely to be used to avoid clean air policies or used in areas in which smoking is restricted.

## 1. Introduction

The IQOS heat-not-burn smoking device from Philip Morris International (PMI) has been available for purchase since 2014, but has only been permitted for sale in the United States after receiving premarket tobacco product application (PMTA) clearance from the United States Food and Drug Administration (FDA) in April 2019 [1]. Following approval for IQOS to be sold in the United States, the FDA granted PMI’s IQOS tobacco heating system a modified risk tobacco product authorization (MRTP) in July 2020. IQOS became the second tobacco product to receive MRTP authorization and the first to receive an “exposure modification” order, permitting marketers to describe the product as presenting a reduced risk of exposure to harmful substances compared to combustible cigarettes [2].

Before receiving MRTP authorization, however, IQOS had been marketed internationally as a cleaner, “cool” tobacco product alternative in Japan [3]. A review of 15 studies found increased awareness and interest in trying heat-not-burn products among young adult smokers and non-smokers [4]. Although few empirical studies have examined users’ perceptions of IQOS, qualitative data from consumers in Switzerland and Japan suggests that consumers may be skeptical about the products’ health benefits compared to combustible cigarettes [3]. Additionally, there is a lack of rigorous studies examining how IQOS may appeal to adolescents. PMI has suggested that there has not been sufficient data assessing whether or not adolescents will find IQOS appealing or perceive it as free of associated health risks [5]. However, adolescents exposed to modified risk tobacco product claims have been shown to demonstrate lower perceptions of risk of tobacco product use and fewer concerns about chemicals in tobacco products [6]. Slovic’s theory of cognitive decision making suggests that experiential, or intuitive, automated cognitive decision making could help explain early research results regarding user risk perceptions of IQOS [7]. Experiential systematic decisions, often rely on heuristic images and associations that can act as cognitive shortcuts in determining risks [7]. Importantly, the novelty of IQOS may help heuristically foster overly optimistic expectations about its health impacts among potential users. Findings from young adults’ reactions to stylized IQOS packaging, including “no smoke” language, suggest that novel product features and a “high-tech” appeal may contribute to decrease users’ assessments of potential harms associated with IQOS use [8]. Additionally, recent research among Taiwanese adolescents found that, while exposure to anti-tobacco educational programs negatively correlated with adolescent e-cigarette or combustible cigarette use, it had no effect on IQOS usage rates [9].

Heated tobacco products (HTPs) are an emerging type of tobacco product engineered differently from electronic cigarettes (e-cigarettes) [10]. Specifically, HTPs and e-cigarettes are both battery-powered devices. Yet, HTPs heat processed tobacco leaves to emit aerosol without combustion [10], while e-cigarettes heat a liquid usually containing nicotine, propylene glycol, glycerin, and flavorings into an aerosol [11]. In August 2016, the FDA started to regulate e-cigarettes, e-liquid, and all related products [12]. The HTPs were not authorized by the FDA until April 2019. The first HTP authorization were for IQOS tobacco heating system, along with the tobacco-containing “Heetsticks”, which included the tobacco, paper, and filter inserted into the IQOS device [1]. In July 2020, the FDA announced further authorization for the marketing of IQOS as “modified risk tobacco products (MRTPs) [1]”. Specifically, the FDA authorized the “reduced exposure” marketing claim to IQOS products, indicating its use results in reduced exposure to harmful and potential harmful toxicants than conventional cigarettes [1]. However, little is known so far about the relative harm between using HTPs and e-cigarette products.

Tracking and assessing real-world conversations and depictions of novel tobacco products as they enter the United States market is one method that can help inform regulatory efforts. By examining unprompted conversations about novel tobacco devices, researchers can provide key information about marketing efforts as well as relevant consumer risk perceptions and health beliefs. One approach that has been utilized to examine perceptions and narratives surrounding new tobacco products is the study of tobacco-related social media content. Previous research has found associations between user discourse of tobacco products on social media [13] as well as engagement with tobacco-related content on social media [14] to increased risk of tobacco use risk. Currently, prior research examining IQOS-related conversations on social media has been very limited. One study analyzed PMI marketing strategies and described a social media campaign in Israel that distributed positive promotional content related to IQOS [15]. Another study looked at IQOS influencer marketing on Instagram in Czech Republic and found that some Czech celebrities presented IQOS as part of aspirational and healthy lifestyle in their Instagram promotional posts [16]. Thus far, no studies of which we are aware have attempted to assess how health impacts of using IQOS have been discussed in a social media environment.

This study aimed to fill this literature gap and categorize Twitter conversations involving IQOS as it became available for purchase in the United States. Findings should contribute to policy targets and communication campaigns by documenting early attitudes and perceptions of an emerging tobacco product.

## 2. Materials and Methods

Data for our study were comprised of original (non-retweet) posts containing the term “IQOS” were collected from 1 January 2020 to 30 June 2020 from Twitter’s Streaming API. These inclusion parameters follow previous research examining brand-specific vaping conversations [17,18]. A six-month data collection timeframe was adopted to balance data collection time and the amount of data that would be feasible on which to perform a qualitative exploratory thematic analysis. There were n = 151,926 unique tweets during this time. Based on previous research that has examined social media content and vaping product conversations, we restricted our analysis to English-language tweets [19]. However, only 2.5% of these tweets were recognized in English (n = 3916). The conversation around IQOS on Twitter was mostly in Japanese. The English sample (n = 3916 tweets) was analyzed by manual content analysis, utilizing a directed qualitative approach [20]. Initially, we developed a codebook informed by themes found in previous tobacco-related research that relied on social media data [21,22]. These themes were supplemented by an iterative examination of 250 IQOS posts in which content that did not fit into initial codes was identified and determined to be either a subtheme of an existing theme or a novel theme [20]. A final coder assessed 50 tweets and made recommendations for refining the code book.

Six main themes were identified through this process including: (1) News (a news story or piece of news information about IQOS or its associated brands, e.g., a market commentary tweet talking about Philip Morris International stocks); (2) Health Claims (a health-related claim about IQOS); (3) Marketing (a marketing claim, e.g., promotes a flavor or point of sale for IQOS), (4) Personal Testimony (personal description of IQOS, e.g., use or opinions about the product), (5) Tobacco Policy (a description of IQOS and some form of United States or international tobacco policy or regulation), and (6) Other (any tweet that does not fit into any other category listed in the codebook, is not in English, or cannot be interpreted). Organizational and personal tweets were determined in an iterative fashion by each coder once themes were identified. In other words, tweets saying “I tried” or “when I used” were coded as having derived from an individual, while tweets describing what “our brand”, a shop or outlet, or with organizational indicators in the username (e.g., “@APNews”) were not coded as having derived from individuals. “News” and “Marketing” themes were also iteratively determined based on the content of the tweet, specifically whether coders interpreted the tweet as attempting to sell/promote a product or to inform in a journalistic fashion.

All main categories except for News and Other were further divided into the following 13 sub-categories on an iterative basis: Health Claims: *combustible* (a claim that IQOS is healthier or less harmful than combustible tobacco), *vaping* (a claim that IQOS is healthier or less harmful than other vaping products), *cessation* (a description of IQOS and quitting either combustible tobacco or vaping); Marketing: *marketing flavor* (marketing of an IQOS flavor to consumers), *product features* (marketing of an IQOS feature to consumers, e.g., design or accessories); Personal Testimony: *sentiments* (any consumer description of IQOS use and positive or negative opinions about the product), *flavor* (a description of flavor in the context of a personal recommendation, thought, or review), *experience* (a description of the experience of using IQOS either favorably or unfavorably, e.g., feeling of smoke, design, build quality), *inquiry* (a consumer inquiry from another source about IQOS use, flavors, policies or to troubleshoot issues with an IQOS product); Tobacco Policy: *United States* (a description of IQOS and some form of the United States tobacco policy or regulation), *international* (a description of IQOS and some form of international policy or regulation), *clean air* (a tweet describes IQOS and a policy specifically regulating the use of tobacco products in certain areas), *flavor bans* (a tweet describes IQOS and a policy specifically regulating or banning flavored tobacco products). Policy tweets were only coded as *United States* or *international* if a specific country or regulatory body was described. All policy-related tweets that did not specifically indicate a country of origin were coded as Tobacco Policy, but not coded into *United States* or *international* subthemes.

Two independent coders coded the first 1000 tweets using the binary system for each category and sub-category (“1” if a specified description was present and “0” if a specified description was absent). The inter-rater reliability (IRR) for all themes and subthemes was calculated after the initial coding; initial IRRs ranged from 72% to 98%. Following previous qualitative content analysis best practices [23], theme and subtheme differences were discussed and coding definitions amended until consensus between the two coders was reached, and the remaining tweets were coded with the final IRR for all categories > 87%. After the coding process was completed, we conducted a descriptive analysis of the categorized tweets and a pairwise co-occurrence analysis to evaluate connections across themes and subthemes (e.g., the proportion of tweets describing personal testimonies that have health claims about IQOS) as well as between the main themes and their corresponding subthemes (e.g., the proportion of marketing tweets that described flavors) to provide a more nuanced content analysis. Co-occurrence analyses were run in R using the tnet packages to construct the co-occurrence matrix and the igraph package to visualize the results. All posts in this dataset were publicly available and anonymized, and all analyses adhered to the terms and conditions, terms of use, and privacy policies of Twitter, and were performed under University of Southern California Institutional Review Board approval. To further protect privacy, posts exemplifying themes are paraphrased; no tweets are reported verbatim in this report.

## 3. Results

### 3.1. Resultant Themes

Table 1 shows the descriptive characteristics of the sample of tweets and sample tweets illustrating each theme. Many of the 3916 tweets in the final corpus were either examples of online **Marketing** (32.3% of corpus) or **Personal Testimonials** related to IQOS use (34.2% of corpus). Over one-third of **Marketing** tweets described flavored “HeetSticks” (34.6% of theme). HeetSticks, or long strips of processed and reformed tobacco which are inserted into the IQOS holder for use, were promoted as offering “Tropical menthol”, or color-based flavored tobacco packs such as “Purple label Heets” which, as one **Marketing** tweet claimed, “may be a tobacco mix with blueberry style, wild herb, and a bit of lotion”. A smaller subset of **Marketing** tweets described specific product features or accessories (5.3% of theme) such as bejeweled cases that could be bought to personalize IQOS systems.

**Personal Testimonials** regarding IQOS were mixed. Of the 1340 tweets that coders identified as a personal testimonial about the product (34.2% of corpus), 659 (16.8% of corpus) were coded as presenting a personal sentiment about the IQOS system. Nearly two-thirds of tweets that described a personal sentiment of the IQOS system were identified as expressing a negative sentiment (n = 413). Negative sentiment tweets were often related to personal experiences associated with the using the device such as foul odor (e.g., “The smell stinks and stains”). Other tweets made negative **Health Claims** associated with using IQOS, rejecting marketing or proclaimed narratives of risk reduction (e.g., “IQOS causes cancer, it’s not harm reduction if [it causes] death”). Nearly half of the tweets describing personal experiences were negative (43.0% of subtheme) (e.g., “…[IQOS] is trash and I have huge problems with it”), while only 31.7% (n = 78) of personal experience subtheme tweets positively described using or encountering IQOS devices (e.g., “I’m [pausing] smoking ahead of my marathon next week. … [I can’t wait] to go back home to my IQOS”).

Positive sentiments (37.3% of theme) about IQOS suggested the product may help current smokers quit using combustible cigarettes or described the process of moving from one combustible alternative to IQOS (e.g., “[My] friendship with e-cigs [has] ended. IQOS is my new best friend now”). Positive **Health Claims** about IQOS in comparison to either vaping or combustible cigarettes was also present in our data. One specific claim mentioned over 40 times in our data suggested IQOS was either 95% or 95 times less health adverse for users than combustible tobacco or vaping.

**Personal Testimonial** tweets in our corpus did not just offer opinions about IQOS products. Individual tweets also sought information from outside sources about the product. **Personal Testimonials** queried either other users (e.g., “IQOS vs Glo—which is better?”) or branded IQOS accounts (7.1%% of corpus) (e.g., “@iqos_support_sa Hello there, where can I find [IQOS Heetsticks] in Saudi Arabia?”) to gain more information about user preferences, IQOS promotions, or availability.

**Tobacco policy** topics were found in a significant number of all tweets (13.9%), split almost evenly between discussions of United States (4.9%) and international (4.4%) tobacco policies. United States policy tweets described how IQOS may benefit from flavored tobacco product bans (13.2% of theme), or queried the US Food and Drug Administration about how IQOS may fit into the current regulatory environment. Nearly 14% of **Tobacco policy** tweets described either how IQOS could be used to circumvent clean air policies or how the devices could be used in areas that do not allow smoking or vaping. Still, other tweets suggested collusion between politicians such as former New York mayor Michael Bloomberg to champion IQOS over other alternative tobacco products (e.g., “… How much money was [given by @InsidePMI] in order to have Mr. Blair speak on [IQOS’s] behalf at the FDA hearing […]?”). International policy tweets either discussed how international regulations may benefit IQOS (e.g., “The EU menthol ban could be [beneficial for] #IQOS […]”), or ways in which IQOS had avoided falling under clean air regulations in international markets (e.g., “Ahead of Japan’s upcoming new indoor smoking restrictions, you can expect to find more #IQOS only [places]”).

Finally, **News** media tweets comprised a significant portion of our data (14.2% of corpus). **News** media tweets about IQOS primarily discussed either health claims about the device (26.8% of theme) or tobacco policy issues facing IQOS (34.6% of theme). About one-third of news media tweets that made health claims about IQOS made direct health impact comparisons between the device and combustible cigarettes (n = 44), while only five news media tweets made comparisons between IQOS and other ENDS products. Policy-related news media tweets were more likely than tweets from all sources to focus on how international policies (45.1% of the news media posts) rather than United States policies (21.2%) may impact IQOS sales or availability.

### 3.2. Thematic Co-Occurrences

As tweets can often describe more than one topic at a time, we also collected data on co-occurrences between categories within the same tweet (see Figure 1). The most frequent co-occurrences of tweets (11.1% of corpus) were observed between **Marketing** tweets and tweets describing IQOS flavors, indicating that advertising flavors available to consumers was an integral part of the IQOS marketing strategy in our data. The second most frequent co-occurrence in our data was observed in **Personal Testimony** tweets in connection with **Tobacco Policy**. Nearly two-thirds (n = 344) of the 545 Tobacco Policy-related tweets in our corpus were written by individuals whose tweet also shared **Personal Testimonials** about using IQOS products.

Additional co-occurrence analyses indicated the presence of just under 100 tweets **Marketing** IQOS by making **Health Claims** about the device. **Health Claim** marketing tweets comprised nearly eight percent of all **Marketing** tweets about IQOS. Describing IQOS use alongside health discussions was more prominent in personal testimonial tweets. Health claims about IQOS were also found in tweets describing personal use of IQOS products. Personal testimonial tweets made **Health Claims** about IQOS often in comparison to other forms of tobacco products. Specifically, these tweets often described health impacts of IQOS products either in comparison to combustible cigarettes (n = 80) or vaping products (n = 14). Although tweets often discussed comparative health impacts of IQOS and other tobacco products, a larger subset of personal testimonial tweets presented co-occurrences with cessation discussions as users discussed IQOS as an avenue to quit smoking/tobacco use (n = 133).

## 4. Discussion

This study categorized English-language discussions about IQOS products on Twitter to investigate how the product has been discussed by English-speaking Twitter users and marketed to potential consumers prior to its availability in the United States market. The majority of tweets in our corpus were either dedicated to marketing IQOS to potential consumers in countries in which the products were currently available or were crafted by individuals who had had some form of experience either using the product or encountering its use.

Our findings suggest that topics discussed on Twitter about IQOS largely conform to previous characterizations of other emerging tobacco products (e.g., ENDS) [19,21,24,25]. Similar to studies of Reddit [9] and Twitter examinations of e-cigarette discussions [26], we found that marketing-related IQOS posts were a key driver of IQOS content in our corpus. Such promotional posts focused on the product’s flavors, which also aligns with previous research [21,27]. Although our co-occurrence analyses demonstrated a disproportionate reliance on selling IQOS based on the flavors available, there were also a significant number of marketing tweets emphasizing potential health benefits for using IQOS products. These findings align with previous research suggesting IQOS marketing may mislead consumers into believing the product is risk-free [5].

The number of personal testimonials describing health benefits of IQOS found in this study was cause for concern. Specifically, the claim repeated over 40 times in our corpus that IQOS was 95 times “more healthy” or “cleaner” than combustible cigarettes or e-cigarettes. This claim, a misrepresentation of findings from a 2015 Public Health England commissioned report [28], about the health impacts of vaping compared to combustible cigarettes has been particularly pervasive in lay discussions of alternative tobacco products. The presence of personal testimonials describing the comparative health benefits of IQOS over other nicotine products adds to the current literature on user perceptions of IQOS. Previous research findings have been mixed, with some research suggesting consumers may find IQOS to be a less-harmful nicotine alternative [29], despite contradictory evidence suggesting consumers may not perceive IQOS as less harmful [3]. Tweets in our corpus often described IQOS as a less harmful alternative to other nicotine products or as a cessation aid, but user perceptions of the actual experience of using IQOS were not as positive. In other words, the majority of reviews of IQOS in our data were negative. These findings offer support to previous qualitative research suggesting consumers may find the experience of using IQOS compares unfavorably with combustible cigarettes [3].

The data from our study also indicate that the authors of English-language tweets about IQOS prior to PMTA approval of the product were written by a highly engaged audience, meaning they often tweeted from a position of personal knowledge about the product and the tobacco market in general. This is evidenced by a significant presence of policy related discussions about the product (both international and domestic), as well as inquiries and reports about how the release of IQOS could impact PMI’s stock performance. Individuals discussing IQOS often had direct experience with the products, either through personal use, or had experienced IQOS users in their daily lives. Although the majority of personal opinions regarding the product were negative, authors of tweets in our data set often expressed nuanced assessments of IQOS products—weighing the negatives associated with the product (e.g., bad smell) against its potential benefits (e.g., potential combustible cigarette cessation tool). Overall, users in our corpus presented a complex narrative portraying IQOS as a product that would likely be healthier than using other nicotine products, but is ultimately hampered by an unsatisfying user experience. These initial findings point to a decidedly mixed opinion about IQOS among individuals discussing the product on Twitter. Some public health scholars have argued that, although heat-not-burn tobacco devices contribute to nicotine addiction as well as other deleterious effects on users’ health, their use by nicotine-dependent smokers in tightly regulated contexts could constitute harm reduction [4,30]. Awareness and usage of these products has also been shown to be growing in the United States as well as internationally, with IQOS being a leading brand in the segment [4,31]. Data collected from our corpus indicates that, overwhelmingly, tweet authors were optimistic about the overall harms associated with IQOS compared to combustible cigarettes, but that tactile and other negative user experiences (e.g., smell) associated with the product may limit uptake of the product among nicotine-dependent smokers.

One of the most frequent themes we identified in our corpus was tweets attempting to persuade users to purchase IQOS. Tweets in our data about marketing IQOS in our corpus focused overwhelmingly on describing the various flavors that were available in international markets. Retailers selling IQOS should not be able to market or sell the various berry or fruit flavored “Heetsticks” that were prominently featured in our corpus in the American market, due to the FDA’s enforcement policy targeting flavored cartridge-based e-cigarettes. However, as menthol flavors are not currently under the same marketing restrictions as other tobacco flavors, future research should anticipate and examine the presence of marketing tweets emphasizing menthol IQOS Heetsticks. Additionally, the conversations in our study were primarily driven by an engaged audience who not only offered their opinions about using IQOS, but also speculated about the product’s impact on PMI’s market standing.

Our study was not without its limitations. This study focused on the text of posts but did not code website links or images attached to posts. Findings may not extend to other time periods or other social media platforms. Posts from this study may not reflect the attitudes from Twitter users with private accounts. Data from this study were comprised of English-language tweets that constituted only about 5% of tweets mentioning IQOS in our dataset. Although our content analysis revealed a wide variety of domestic and international topics, we were not able to make inferences about Twitter conversations about IQOS among non-English speaking Twitter users. It is important to note that, although it was not possible to determine whether tweets in our corpus were composed by adolescents or adults, nearly one-third of American teens utilize Twitter [20]. Although our coders determined during analyzing the data whether each tweet was most likely from an individual or organization, we did not analyze specific usernames of the tweets’ authors to avoid possible sparsity in username category and since the main focus of this project was content analysis in two broadly defined tweet groups: user-generated and promotional.

Results from our analyses should provide guidance for future research. Future analyses of IQOS tweets can expand upon these findings by investigating the extent to which themes discussed by an engaged audience in our data compare to social media discussions after IQOS was made widely available in the United States. As IQOS is the first ENDS product to achieve an FDA modified exposure risk classification, future research into IQOS digital marketing should explore whether the health marketing claims about the product align with the specific exposure risk language deemed acceptable within the exposure risk classification [1]. Future research should extend these findings by examining non-English tweets about IQOS product, investigating post-modified exposure risk marketing and discussions, and also investigate the potential presence of bots or spam accounts in the generation of personal testimonials about IQOS, which could serve to shape perceptions regarding the overall risks associated with the product. Future research should consider the potential negative impact exposure to marketing and positive personal depictions of IQOS use may have on adolescent curiosity, harm perceptions, and experimentation with IQOS. Future studies should focus on analyzing non-English IQOS-related tweets, since these tobacco-products are more popular abroad, especially in Eastern Europe and Asia, compared to the United States.

## 5. Conclusions

In this study, we found most of the Twitter discussions in our data focused either on marketing IQOS to potential consumers or personal testimonials about the product. A small, but present number of marketing tweets made specific health claims about IQOS either as a cessation tool or as a less harmful alternative to other tobacco products. We also found active discussions about IQOS and tobacco policy, suggesting individuals discussing IQOS on Twitter are likely engaged users or commenters. Although IQOS may reduce user exposure to harmful chemicals relative to combustible cigarettes, its use could still increase tobacco-related morbidity and mortality.

## Figures and Tables

**Figure 1 ijerph-18-04836-f001:**
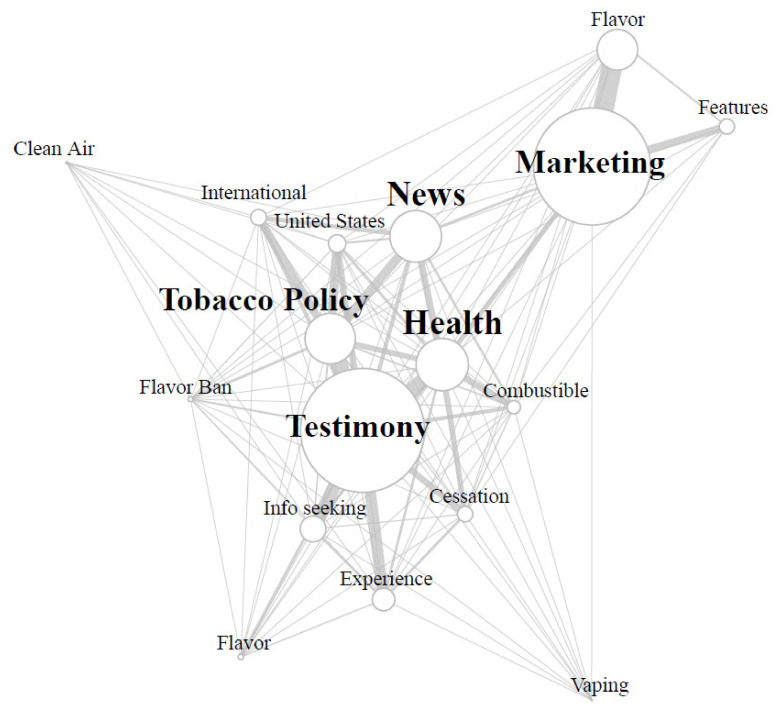
Co-occurrences of themes in corpus tweets. Note: Size of circles represents tweet theme frequencies. Line size and position of circles represent frequency of co-occurrences. Larger lines indicate greater co-occurrence, circles closer to one another were more likely to co-occur.

**Table 1 ijerph-18-04836-t001:** IQOS tweet categorical descriptives and representative tweets (N = 3916).

Topic	N	% Corpus	% Theme/Subtheme	Example
**News**	558	14.2	-	In Ukraine, #IQOS has likely been helped […] by the #IQOSfriendly program[.] […Bars] and restaurants permit customers to use IQOS indoors. […] https://xxxxxx
**Health: Overall**	566	14.5	-	-
*Health: Combustible*	143	3.7	25.3	@xxx [… I] think it’s called iqos. [It’s] technically more like smoking but said to be safer than cigarettes. [I] don’t really know […]
*Health: Vaping*	21	0.5	3.7	@xxx Now, they’re producing and promoting @¡qos[…] that heats tobacco, like vaping […] https://xxxxxxx They say it’s healthier than cigarettes and vaping, but I doubt it.
*Health: Cessation*	167	4.3	29.5	@xxx I switched to IQOS four months ago [following] 37 years of smoking […]
**Marketing: Overall**	1265	32.3	-	-
*Marketing: Flavor*	438	11.2	34.6	NEW 400sticks Marlboro iQOS Heat Sticks Tropical menthol https://xxxxx
*Marketing: Features*	165	4.2	13.0	#smoke #shisha Solid Color Case for IQOS 2.4 Plus https://xxxxxx
**Testimony: Overall**	1340	34.2	-	-
*Testimony: Sentiment (+)*	246	6.3	18.4	@xxx_ I’ve been using Iqos device since November and loving it. I don’t smell like a cig and I can still enjoy the feeling of a puff
*Testimony: Sentiment (−)*	413	10.5	30.8	hi ur juul/iqos dont smell better” than cigarettes pls stop using those indoors ty”
*Testimony: Flavor*	58	1.5	4.3	@xxx my friend has convinced me to switch to iqos because [they kept] menthol
*Testimony: Experience*	244	6.2	18.2	@xxxx I’ve been using Iqos device since November and loving it. I don’t smell like a cig and [it feels similar to combustible cigarettes]
*Testimony: Info seek*	279	7.1	20.8	@iqos_support_ca [The red light is on] in my IQOS What can I do[?]
**Tob. Policy: Overall**	545	13.9	-	-
*Tob. Policy: United States*	190	4.9	34.9	Smoking, Vaping Ban Now In Effect In Atlanta, Hartfield-Jackson https://xx:xxx
*Tob. Policy: International*	171	4.4	31.4	#IQOS, #glo and othe heated tobacco products [will] benefit from Japan’s upcoming smoking ban, which [permits] areas for #HeatedTobacco in bars and restaurants. https://xxx
*Tob. Policy: Clean air*	21	0.5	3.9	Ahead of Japan’s upcoming new indoor smoking restrictions, you can expect to find more #IQOS only [places]
*Tob. Policy: Flavor bans*	54	1.4	10.0	#IQOS gearing up in the UK ahead of #MentholBan https://xxx
**Other**	795	20.3	-	-

Note: Percentages do not add up to 100% as themes were allowed to co-occur. Tweets are paraphrased to preserve anonymity.

## Data Availability

Data sharing not applicable.

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
