# Peer review of "Categorizing IQOS-Related Twitter Discussions"

_ijerph, 2021, doi:10.3390/ijerph18094836_

Round 1
Reviewer 1 Report
Dear Author(s),
Many thanks for the opportunity to review this paper.
It is a very interesting research with the aim to analyzing product IQOS as an alternative and ‘cool’ tobacco. The research has been developed through content analysis on Twitter.
According to the current version of the paper, I want to suggest some changes to improve the readability and critical contribution of this valuable research.
1. I suggest to the author(s) to adding a section entitled “Theoretical framework".
2. I encourage the author(s) to expand the conclusions.
3. The paper could be strengthened from underlining how it positions it-selves in relation to the state of the art. What is already covered in the literature and what needs to be filled in to understand the relationship between health, tobacco, product IQOS and its discourse on social media (Twitter). In this sense, the literature review regarding these issues is non-existent.
Please take these examples as an invitation to create a theoretical framework of your interesting research:
a) Max, W. B., Sung, H. Y., Lightwood, J., Wang, Y., & Yao, T. (2018). Modelling the impact of a new tobacco product: review of Philip Morris International’s Population Health Impact Model as applied to the IQOS heated tobacco product. Tobacco control, 27(Suppl 1), s82-s86.
b) Hair, E. C., Bennett, M., Sheen, E., Cantrell, J., Briggs, J., Fenn, Z., ... & Vallone, D. (2018). Examining perceptions about IQOS heated tobacco product: consumer studies in Japan and Switzerland. Tobacco control, 27(Suppl 1), s70-s73.
c) Kim, M. (2018). Philip Morris International introduces new heat-not-burn product, IQOS, in South Korea. Tobacco control, 27(e1), e76-e78.
d) Farsalinos, K. E., Yannovits, N., Sarri, T., Voudris, V., Poulas, K., & Leischow, S. J. (2018). Carbonyl emissions from a novel heated tobacco product (IQOS): comparison with an e‐cigarette and a tobacco cigarette. Addiction, 113(11), 2099-2106.
e) Lee, J. G., Blanchflower, T. M., O'Brien, K. F., Averett, P. E., Cofie, L. E., & Gregory, K. R. (2019). Evolving IQOS packaging designs change perceptions of product appeal, uniqueness, quality and safety: a randomised experiment, 2018, USA. Tobacco control, 28(e1), e52-e55.
Reviewer 2 Report
Abstract:
Please specify whether the Tweets selected for the study were linked to a specific geographic location (i.e. whether these were selected from the US only)
Please specify what this study adds.
Introduction:
Please add a short paragraph on how IQOS differ from e-cigarettes and how regulations of IQOS differ from those of e-cigarettes- this will help to understand the context of non-combustible tobacco product regulatory context in the US
Please add information on prevalence of smoking, IQOS use and e-cigarette use if possible. In the current version essential context information is missing.
Methods:
Please justify the timeline for selecting tweets & explain whether when selecting the tweets language was the only criteria
How is ‘news’ theme separated from ‘marketing ‘theme or 'health' theme (based on what’s reported in the results it seems that news theme covers aspects similar to those covered in health claim theme)?
Please explain in your methods/ results how you were able to assign a tweet to either US or international tobacco policy (for example, there was menthol ban implemented in Europe in May 2020)?
Please also clarify whether you in any way explored who was publishing these tweets i.e. whether these were companies or individuals. If you did not look at the profile of tweeters please add a brief explanation on why that was not considered.
Discussion:
Discussion claims that the products indicate how the products might be understood in the US- but it seems to contradict the fact that these tweets come from around the world and refer to different regulatory settings
As there are countries where IQOS have been on the market for a while, like Japan, it would be important to include any evidence on whether IQOS have been beneficial/ harmful to public health and how the findings reported in this study relate to potential effects on use of tobacco or non-tobacco nicotine products
Issues related to generalizability need to be discussed -analysis covering only 2.5% tweets in English, and if these cover the entire world and not one specific region- how do they relate to various regulatory environments and whether these can be considered to be representative of iqos debate needs to discussed
Methods section suggest that 2.5% of tweets on IQOS were in English while discussion suggests that it was 5%- please clarify
Statement in Conclusions section: “Although IQOS may 305 reduce user exposure to harmful chemicals relative to combustible cigarettes, its use could 306 still increase tobacco-related morbidity and mortality.” Isn’t in any way related to what has been reported in the study and should be removed/ rephrased
Round 2
Reviewer 1 Report
Dear Authors,
Thank you for submitting the revised version of your paper to the International Journal of Environmental Research and Public Health. I'm very happy to announce you that, in my opinion, the paper has been considerably improved.
Reviewer 2 Report
No further comments